# PopART-IBM, a highly efficient stochastic individual-based simulation model of generalised HIV epidemics developed in the context of the HPTN 071 (PopART) trial

Michael Pickles[1]*, Anne Cori[1º], William J. M. Probert[2º], Rafael Sauter[2º], Robert Hinch[2], Sarah Fidler[3,4], Helen Ayles[5,6], Peter Bock[7], Deborah Donnell[8], Ethan Wilson[8], Estelle Piwowar-Manning[9], Sian Floyd[10], Richard J. Hayes[10], Christophe Fraser[2], HPTN 071 (PopART) Study Team

1 Medical Research Council Centre for Global Infectious Disease Analysis, School of Public Health, Imperial College London, London, United Kingdom, 2 Big Data Institute, Li Ka Shing Centre for Health Information and Discovery, Nuffield Department of Medicine, University of Oxford, Oxford, United Kingdom, 3 Department of Infectious Disease, Imperial College London, London, United Kingdom, 4 Imperial College NIHR BRC, London, United Kingdom, 5 Zambart, School of Public Health, University of Zambia, Ridgeway Campus, Lusaka, Zambia, 6 Department of Clinical Research, London School of Hygiene and Tropical Medicine, London, United Kingdom, 7 Desmond Tutu TB Centre, Department of Paediatrics and Child Health, Stellenbosch University, Cape Town, South Africa, 8 Vaccine and Infectious Disease Division, Fred Hutchinson Cancer Research Center, Seattle, Washington, United States of America, 9 Department of Pathology, Johns Hopkins University School of Medicine, Baltimore, Maryland United States of America, 10 Department of Infectious Disease Epidemiology, London School of Hygiene and Tropical Medicine, London, United Kingdom

º These authors contributed equally to this work.
* m.pickles@imperial.ac.uk

**Data Availability Statement:** All files (model code and parameter files) are available via a doi stated in

## Abstract

Mathematical models are powerful tools in HIV epidemiology, producing quantitative projections of key indicators such as HIV incidence and prevalence. In order to improve the accuracy of predictions, such models need to incorporate a number of behavioural and biological heterogeneities, especially those related to the sexual network within which HIV transmission occurs. An individual-based model, which explicitly models sexual partnerships, is thus often the most natural type of model to choose. In this paper we present PopART-IBM, a computationally efficient individual-based model capable of simulating 50 years of an HIV epidemic in a large, high-prevalence community in under a minute. We show how the model calibrates within a Bayesian inference framework to detailed age- and sex-stratified data from multiple sources on HIV prevalence, awareness of HIV status, ART status, and viral suppression for an HPTN 071 (PopART) study community in Zambia, and present future projections of HIV prevalence and incidence for this community in the absence of trial intervention.

the methods (https://doi.org/10.5281/zenodo.3522848).

**Funding:** RH, SF, PB, HA and CF received funding as part of the HPTN 071 (PopART) trial, which was supported by funding from the National Institute of Allergy and Infectious Diseases (NIAID, https://www.niaid.nih.gov/) under Cooperative Agreements UM1-AI068619, UM1-AI068617, and UM1-AI068613, with funding from the U.S. President's Emergency Plan for AIDS Relief (PEPFAR). Additional funding was provided to RH, SF, HA, PB and CF by the International Initiative for Impact Evaluation (3ie, https://www.3ieimpact.org/) with support from the Bill & Melinda Gates Foundation (https://www.gatesfoundation.org/), as well as by NIAID, the National Institute on Drug Abuse (NIDA, https://www.drugabuse.gov/) and the National Institute of Mental Health (NIMH), all part of the U.S. National Institutes of Health (NIH). MP and AC acknowledge funding from the MRC Centre for Global Infectious Disease Analysis (reference MR/R015600/1), jointly funded by the UK Medical Research Council (MRC) and the UK Foreign, Commonwealth & Development Office (FCDO), under the MRC/FCDO Concordat agreement and is also part of the EDCTP2 programme supported by the European Union. The funders had no role in study design, data collection and analysis, decision to publish, or preparation of the manuscript.

**Competing interests:** The authors have declared that no competing interests exist.

## Author summary

In this paper we present PopART-IBM, an individual-based model used to simulate HIV transmission in communities in high prevalence settings. We show that PopART-IBM can simulate transmission over a span of decades in a large community in less than a minute. This computational efficiency allows us to calibrate the model within an inference framework, and we show an illustrative example of calibration using an adaptive population Monte Carlo Approximate Bayesian Computation algorithm for a community in Zambia that was part of the HPTN-071 (PopART) trial. We compare the detailed model output to real-world data collected during the trial from this community. Finally, we project how the HIV epidemic would have changed over time in this community if no intervention from the trial had occurred.

## Introduction

Mathematical models and simulations are key tools for evidence synthesis across scales of observation, for interpretation, extrapolation, policy planning, cost-effectiveness evaluation and long-term predictions. In HIV epidemiology, models have provided quantitative insights into how acute infection [1], heterogeneous patterns of sexual behaviour [2–4], heterogeneous uptake of interventions and other factors affect the spread of HIV in populations, both in general populations and amongst key populations [5]. Models are used to predict the potential impact of different prevention interventions [6,7], and in the design and evaluation of trials and interventions [8–10].

Increasing amounts of data are available through routine monitoring surveys [11,12] trials (e.g. [13]), and cohort studies [14–16], and they often span multiple time-points. These datasets contain information on risk and prevention behaviours, including detailed questions on sexual partnerships, as well as HIV prevalence. Phylogenetic studies provide new quantitative insights into transmission dynamics [17]. To synthesise these diverse data, there is a need for a new generation of mathematical models that are sufficiently granular (i.e. where the individual is the epidemiological unit that is simulated), and that integrate up-to-date understanding of HIV epidemiology, to produce credible and accurate predictions. Such models also need to represent the multi-faceted activities currently used in HIV combination prevention packages, where a diverse set of interventions are delivered in an integrated manner.

Given the task of developing a model of sufficient complexity to integrate diverse data sources, and that accurately represents the overlapping heterogeneities inherent in HIV population dynamics, an individual-based model (IBM) may be the most parsimonious representation of the epidemic and interventions, compared to compartmental models that are specified in terms of differential equations. Specifically, there are at least three processes that occur on very similar time-scales to the HIV epidemic, and that affect its trajectory: firstly, the population turns over and its demographics change due to births, migration and ageing; secondly, individuals form and dissolve sexual partnerships, sometimes concurrent; and thirdly, infected individuals progress through stages of disease and treatment. These overlapping processes are difficult to fully represent in a parsimonious model consisting of differential equations, and the resulting models may be as or more unwieldy as a typical IBM.

However, it remains challenging to code a complex IBM of HIV transmission that can be run in a computationally efficient manner. Unless each realisation of the simulated epidemic can be obtained quickly, it becomes difficult (and in many cases impossible) to explore a wide range of possible combinations of parameter values, and so obtain confidence in the generality

and robustness of findings from the model. Models are typically paired with a calibration framework when deployed. It is therefore important to not only present a model in isolation but to demonstrate how a complex IBM can realistically be parameterised using setting-specific data and calibrated using an established statistical inference framework.

In this article we present the PopART-IBM, a fast and flexible IBM of HIV transmission. The IBM was designed to model the HPTN 071 (PopART) trial conducted in communities in Zambia and South Africa with an HIV prevalence of >10% of the general population [18]; however the model is intended as a flexible model of generalized HIV epidemics in populations where the primary mode of spread is sexual transmission in heterosexual partnerships.

We describe here the design of the model, and demonstrate its computational efficiency. While the focus of this paper is the model itself, we illustrate the use of the model by showing, in detail, a parameterization to a community from the HPTN 071 (PopART) trial. We show how the model is calibrated, via a Bayesian statistical framework, to a mixture of Demographic Health Survey (DHS) and trial data, including measures of knowledge of HIV serostatus, being on antiretroviral therapy (ART) and viral suppression, as well as HIV prevalence stratified by age and sex. Limitations and future work are described in the discussion.

## Methods

The PopART-IBM is an open-source discrete-time stochastic individual-based model coded in the C programming language. We have structured the code of the model in a modular fashion so that components specific to the HPTN 071 trial, such as the trial intervention or the Population Cohort sample, are coded using separate data structures and in separate files. This allows components to be removed or modified in a straight-forward manner, allowing others to contribute to the model and add components (including other interventions) that are not there at present.

The model is also simple to modify for use with other calibration methods. We first describe the model structure, followed by outlining the data sources used to parameterise and calibrate the model in this study. Finally, we outline how the model is calibrated. Full details of the model structure, including parameters and algorithms, as well as key validation steps, are described in S1 Appendix. The simulation code is available at https://github.com/BDI-pathogens/POPART-IBM under the GNU General Public License 3.0, and version 1.0.0 described in this paper is available at https://doi.org/10.5281/zenodo.3522848.

Fig 1 shows the key components of the model.

### Design principles

The design principles were: first, parsimony, namely only including processes for which we had data, had evidence from earlier work that they were important determinants of population dynamics, or needed to be included to evaluate the PopART intervention; second, use a compiled language (the C programming language), to reduce simulation time, and third, extensive use of analytically derived waiting times instead of next-step event simulation. Parsimony was evaluated subjectively in consultation with the whole trial study team. Compared to next-step simulations, the waiting time approach produces faster simulations, but makes for more complex code and increases memory use.

### Model structure

**Spatial structure.**    We chose a meta-population structure, whereby the population can be divided into one or more spatial patches. Individuals form sexual partnerships within and between patches, but do not migrate between patches. In this study we present results based

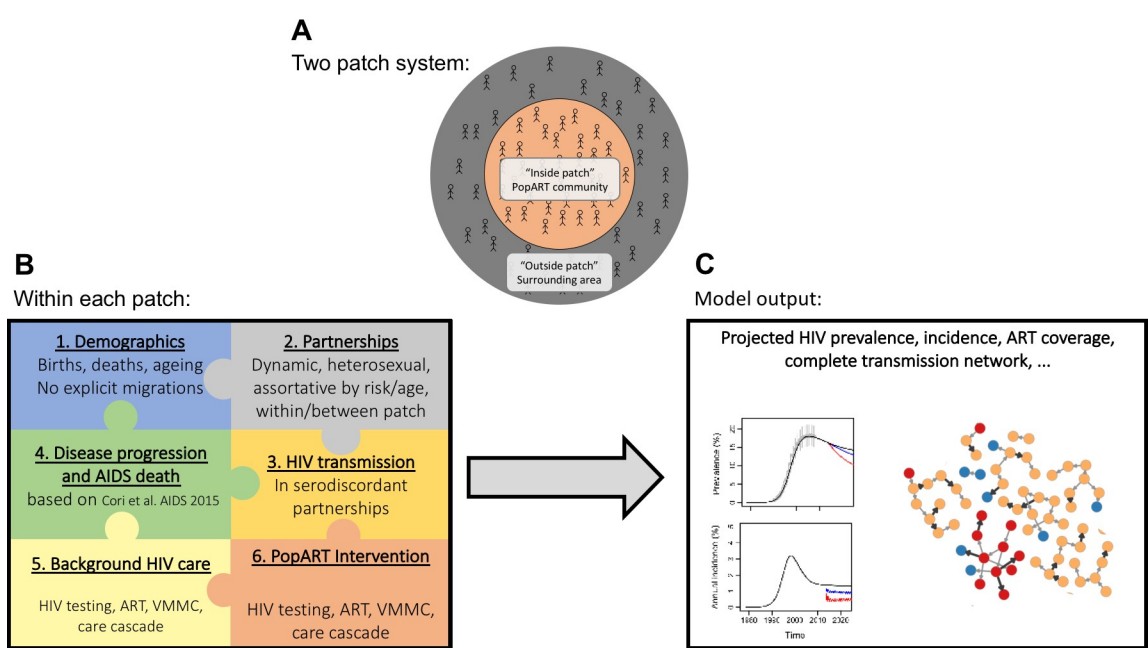

**Fig 1. Overview of the structure of the codebase of the PopART-IBM model.** A) a metapopulation structure for modelling the PopART community (inside patch) and a surrounding area (outside patch); B) within each patch the codebase is partitioned into several processes related to demographics, sexual partnerships, HIV transmission and disease progression, and interventions; C) the nature of the model being individual-based allows detailed output from the model including the complete HIV transmission network.

on two spatial patches, designed to represent the study community (patch 0) and its proximate surrounding area (patch 1) (see Fig 1 left). Individuals may interact by having sexual partnerships with individuals in any patch. The aim of this particular structure is to study the extent to which the impact of an intervention offered to a defined population is modified by partnerships with individuals not benefiting from the intervention. It was parameterised using survey data on where individuals' partners lived and how long this partnership lasted. It could easily be modified to represent more generic spatial structure.

**Demographics.** Births and deaths from non-HIV related causes are modelled based on country-specific age- and time-varying projections from the UN Population Division, as described in S1 Appendix. Mother-to-child transmission of HIV is not modelled: all individuals are HIV-negative at birth. Individuals become available to form sexual partnerships when they turn 14, assumed as a lower age limit for sexual activity. (The actual age of forming the first sexual partnership is an outcome of the partnership formation and dissolution algorithm, and will vary across individuals and over time.)

**Partnerships.** Individuals are divided into three levels of sexual activity ('low'/'medium'/ 'high'). This is assigned when an individual enters the model, and it does not change during an individual's lifetime. The sexual activity level determines the maximum number of concurrent partners they have, which is similarly fixed over their lifetime, and the rate at which they form new partnerships. (The actual number of partnerships is an outcome of the partnership formation and dissolution algorithm, and will vary across individuals and over time). The rate of partnership formation varies by age, so that even individuals with a 'high' level of sexual activity will eventually have fewer partners as they get older. The model simulates partnership formation and dissolution for several decades before HIV is introduced to allow the dynamics of partnership formation to reach a stable state.

After turning 14, individuals may form new heterosexual partnerships throughout the remainder of their lives, at age- and sexual activity level-specific rates, up to their maximum number of concurrent partners at any given time. Key parameters related to partnership formation and dissolution are provided in Table 1. We assume independence between the effects of age and sexual activity level on the rate of partnership formation. Partnerships are formed

**Table 1. Key partnership-related parameters used in PopART-IBM.**

| Parameter | Value | | Notes |
|---|---|---|---|
| | **Zambia** | **South Africa** | |
| Risk assortativity, $\chi$ | 0.05–0.95 | | Large range to reflect uncertainty |
| Proportion compromise from men, $\theta$ | 0.01–0.5 | | Assumption that women underreport more than men. |
| Annual within-community partnership formation rates for low activity men by age group ($c_a^{m,low}$): | | | |
| 13–17 | 0.0174 | 0.0506 | No PC data for 13–17 yo, assumed 50% less than 18–22 in this age group. Age groups from 18–49 informed from PC0 analysis. No PC data available for 50–59 and 60–79 age groups so assume 50% decline in each age group. |
| 18–22 | 0.0348 | 0.1011 | |
| 23–29 | 0.0409 | 0.1615 | |
| 30–39 | 0.0251 | 0.0646 | |
| 40–49 | 0.0243 | 0.0505 | |
| 50–59 | 0.0051 | 0.0152 | |
| 60–79 | 0.0025 | 0.0076 | |
| Annual within-community partnership formation rates for low activity women by age group ($c_a^{f,low}$): | | | |
| 13–17 | 0.0089 | 0.0260 | |
| 18–22 | 0.0177 | 0.0520 | |
| 23–29 | 0.0147 | 0.0286 | |
| 30–39 | 0.0102 | 0.0303 | |
| 40–49 | 0.0131 | 0.0391 | |
| 50–59 | 0.0065 | 0.0196 | |
| 60–79 | 0.0033 | 0.0098 | |
| Relative number of partnerships by sexual risk group ($\delta^r$): | | | |
| Low risk | 1.0 | 1.0 | |
| Medium risk | 8.89 | 2.06 | |
| High risk | 22.4 | 6.38 | |
| Multiplier to account for mis-reporting of number of sexual partners | 0.5–4.0 | 0.5–4.0 | 1.0 means that people report the correct number of partners, > 1 means people are under-reporting. Lower range taken to ensure 1 is sampled well and to allow over-reporting. |
| Relative rate of formation of partnerships between patches compared to within patches | 0.562 | 0.665 | PC0 analysis. |
| Unscaled duration of partnerships within patch between individuals of a particular sexual risk group (years): | | | |
| Low risk | 15.4 | 9.80 | PC0 data |
| Medium risk | 6.1 | 6.99 | |
| High risk | 3.8 | 4.16 | |
| Multiplier for scaling duration of all partnerships | 1.0–2.0 | 1.0–2.0 | Assumption |
| Multiplier for scaling duration of partnerships between patches, compared to within patches | 0.456 | 0.575 | PC0 data |
| Maximum number of concurrent partners by risk group: | | | |
| Low risk | 1 | | Over 90% of PC reported 0 or 1 partners in the last year. Values for medium and high risk are assumptions. |
| Medium risk | 3 | | |
| High risk | 10 | | |

according to an age-mixing matrix (see Tables I-L in S1 Appendix), while mixing between sexual activity levels is governed by an assortativity parameter, $\chi$, allowing mixing to vary from proportionate ($\chi = 0$) to fully assortative ($\chi = 1$) ($\chi$ estimated within the calibration framework). When the model is run with multiple spatial patches, individuals have a lower probability of choosing a partner outside their own patch.

Within an individual-based model, the process of generating a sexual network can be complex. While preferences for mixing by age in partnerships are specified by the age mixing matrix (see in Section S3.3.1), the actual number of partnerships formed between age groups must balance. The algorithm for calculating such balancing first calculates the 'desired' number of partnerships between women ($g = f$) and men ($g = m$) of age group $a$ and risk group $r$ to those of age group $a^*$ and risk group $r^*$, denoted $S^g_{(a,r)(a*,r*)}$. Within the model, this quantity is calculated as

$$S^g_{(a,r)(a*,r*)}(t) = N^g_{a,r}(t)c^{g,low}_a \delta^r p^g_{age}(a, a^*)p^g_{risk}(r, r^*)(t)dt$$

where $N^g_{a^*,r^*}(t)$ is the number of individuals of sex $g$ in age group $a^*$ and risk group $r^*$ at time $t$, $c^{g,low}_a$ is annual partnership formation rate for individuals of sex $g$ in age group $a$ and low activity level group, $\delta^r$ is the relative number of partnerships in risk group $r$ (compared to low), $p^g_{age}(a, a^*)$ are age-stratified mixing values for sex $g$ and from age group $a$ to age group $a^*$, and $dt$ is the duration of a timestep in the model (1/48 year). The matrix of values $p^g_{risk}(r, r^*)(t)$ is the distribution over sexual activity levels ($r$) of desired partners of an individual of sex $g$ of sexual activity level $r^*$ at time $t$, and defined as follows:

$$p^g_{risk}(r, r^*) = \begin{cases} \chi + (1 - \chi)P^{\hat{g}}_r, & \text{if } r = r^* \\ (1 - \chi)P^{\hat{g}}_r, & otherwise \end{cases}$$

where $\hat{g}$ is men if g is women and vice versa; $P^{\hat{g}}_r$ is the proportion of the population of gender $\hat{g}$ which is in activity level $r$ at time $t$. If the 'desired' number of partnerships are not the same for both sexes, an adjusted number of partnerships, calculated as the weighted arithmetic mean between the two with weighting factor $\theta$, is calculated:

$$T_{(a,r),(a*,r*)}(t) = (1 - \theta)S^f_{(a,r),(a*,r*)}(t) + \theta S^m_{(a*,r*),(a,r)}(t)$$

A more detailed description of the algorithm for partnership formation and dissolution is provided in S1 Appendix.

Partnerships last a finite random time, dissolving at a rate that depends on the sexual activity levels of the partners, and whether they reside in the same patch or not (distant relationships last less long). The sexual partnership process is parameterized based on answers to sexual behaviour surveys asked of participants in the community, including the age, residency and duration of recent partnerships, and life-time number of partners at different ages. This determines all parameters except the risk assortativity parameter, $\chi$, which is currently estimated within the calibration framework. Key partnership-related parameters are presented in Table 1, and further details are discussed in S1 Appendix.

**HIV transmission.** At each timestep in the model, HIV transmission occurs within serodiscordant couples, with a probability dependent on the sex of the seronegative partner, and the CD4 count, set-point viral load (SPVL) and antiretroviral therapy (ART) status of the seropositive partner, as well as the circumcision status of the seronegative partner if male. Heterogeneity across the natural history of infection for an individual is thus incorporated via changes by CD4 stage, while heterogeneity between individuals is captured by differences in SPVL and ART status. The relative transmission rates are determined from published

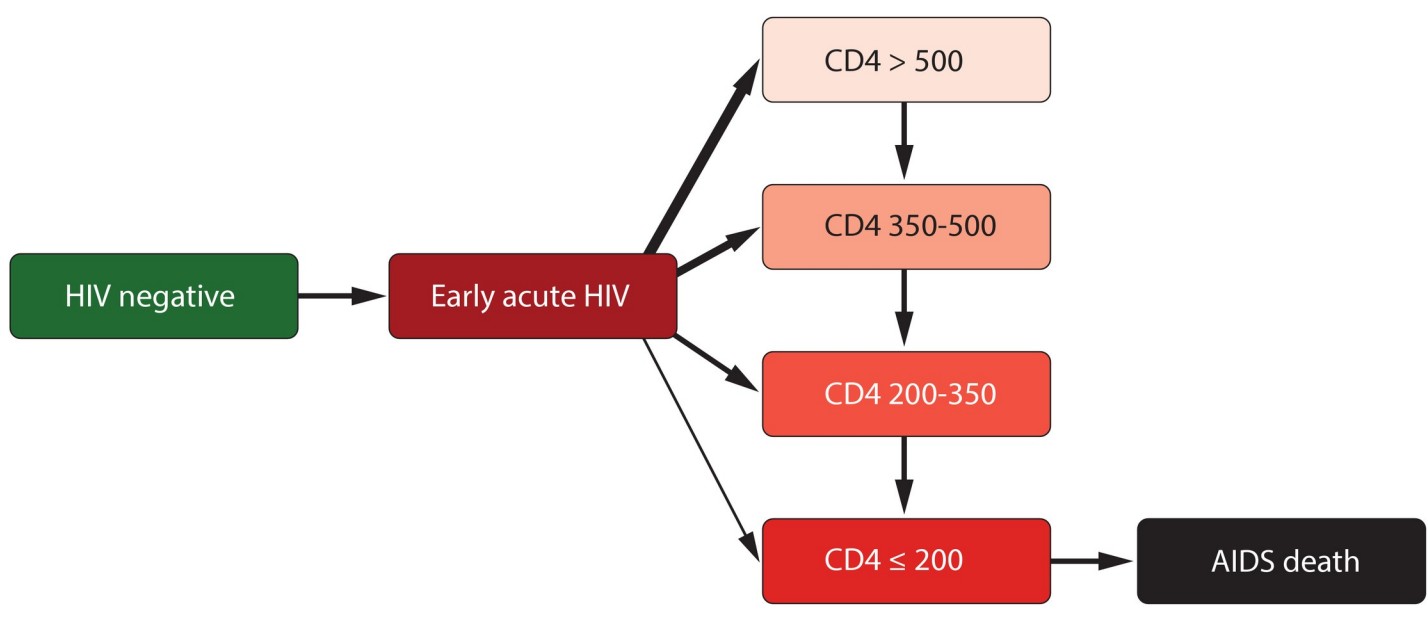

**Fig 2. Schematic of HIV progression in the model in the absence of ART.**

literature on HIV transmission in serodiscordant couples in sub Saharan Africa [1,8,19–22]. HIV transmission rates are reduced when the partners reside in different patches, to reflect the fact that longer-distance partnerships may have lower average coital frequency. HIV transmission is described in more detail in S1 Appendix.

**Disease progression and AIDS.** Upon HIV infection, an individual enters acute and early HIV infection (AEHI), and is assigned a SPVL, which affects both their infectivity and disease progression (Fig 2). After AEHI they enter a CD4 category drawn at random to be one of >500, 350–500, 200–350, ≤200 cells/mm$^3$ with the probability dependent on their SPVL [23]. In the absence of ART and natural death, they then progress sequentially through lower CD4 categories at a rate depending on SPVL which is estimated from long-term patient cohorts [23], until AIDS-related death (Fig K in S1 Appendix). See S1 Appendix for more details on HIV disease progression and AIDS.

**Healthcare and prevention.** In the HPTN 071 (PopART) trial, household-based testing is carried out by community health workers, known as CHiPs (community HIV care providers) in trial Arms A and B (Arm C is the control arm of the trial). The trial intervention package consists of: 1) additional HIV testing through CHiPs teams; 2) enhanced linkage to care; 3) immediate ART regardless of CD4 count in Arm A communities (Arm B communities follow national guidelines, including any changes to guidelines over time); 4) uptake of voluntary male medical circumcision (VMMC) that is potentially differentiated from the background cascade to reflect VMMC counselling as part of the CHiPs intervention. PopART-IBM specifically models the household-based testing using age-/sex-stratified coverage derived from trial data (see S1 Appendix for further details on simulation of the CHiPs intervention) and such a process may be applied to interventions that are delivered based upon age and sex such as PrEP, or vaccination (for modelling of infectious diseases that have a vaccine).

The processes related to HIV testing, linkage to care and voluntary male medical circumcision (VMMC) can occur through multiple channels in the model. For example, in the context of the HPTN 071 (PopART) trial there are 'background' and 'intervention' HIV testing services representing respectively existing testing services through clinics, and PopART-related

services added through the Community HIV-care Providers (CHiPs) teams and other routes [24]. Individuals may test multiple times, through a mixture of background and intervention channels.

Upon receiving a positive test result, individuals can proceed to different stages of the treatment cascade (engaged in care, on antiretroviral therapy (ART), on ART and virally suppressed) or they may drop out at different stages of the cascade. Individuals can start ART if eligible according to local treatment guidelines at that time. Individuals who are not initially eligible may stay in care, until they become eligible either through declines in their CD4+ cell counts or changes in guidelines. Once an individual reaches a CD4+ cell count below 200 cells/mm$^3$, even if unaware of their status, they are assumed to have a chance of seeking medical care because of AIDS-related symptoms, and hence initiating ART. Individuals who start ART may or may not become virally suppressed after an initial period, and can also stop treatment. A full representation of the HIV testing and treatment cascade is presented in S1 Appendix Fig K and further described in S1 Appendix.

A fraction of men are traditionally circumcised during childhood. When undergoing HIV testing in the model, men who test HIV-negative who are not already circumcised may accept referral for voluntary male medical circumcision (VMMC) with a given probability. Only medically circumcised men have a reduced susceptibility to HIV reflecting data from HPTN 071 [25]. All intervention parameters were determined from PopART process data where possible (see calibration below). Male circumcision is described in detail in S1 Appendix

**Notable structural omissions.** In line with our parsimony approach, PopART-IBM does not include many factors included in other models. The model does not include transmission outside of heterosexual partnerships, or in any other way allow for key populations. The model does not explicitly consider migration, as this would require a different approach to parameterizing cross-patch partnerships. The model does not include viral strain structure, such as required to model the spread of antiviral resistance or changing virulence. The model does not currently differentiate between ART regimens. Due to lack of historical data on the communities of interest, we do not model condom use, though we do allow the net transmission rate to be a free parameter (only relative rates are determined from the literature.) We do not model coinfections, including those such as HSV-2 that modify HIV transmission rates. The current model is well suited for generalised heterosexual epidemics with HIV prevalence >10%, as found in the study communities. It is less well suited for epidemics characterised by low overall prevalence and high prevalence among key populations. We plan to address these omissions in future work, where data are available, and where the structural changes are likely to affect predicted interventions. The modular approach in which the model is coded also allows for these components to be added by other researchers using the model.

**Model validation.** The model was constructed in a manner to minimise the risk of introducing errors in the code and ensure the model code is a valid representation of the described processes. The model was built in a modular fashion, splitting the code into related processes (such as HIV transmission, partnership formation, interventions, and so on), and testing of individual components were performed when new processes were added. The Valgrind tool was regularly used to check for any issues with computer memory management. Functions check the input arguments to the model are within plausible ranges, and several checks occur throughout the body of the code so that individuals with implausible event histories (such as becoming HIV positive outside a serodiscordant partnership) will terminate the model and highlight the implausible event. During development, quantities within the model (such as age-stratified prevalence) were calculated from multiple data structures (internal to the model) and compared during runtime. Model output has been reviewed by the PopART modelling team for consistency with observed data. An automated testing framework in Python (pytest)

has been paired with the model to ensure that parameter values lead to expected outputs from the model, and to check the concordance of metrics that are outputted across different files that the model produces (for instance, checking that sex-stratified prevalence is concordant when calculated with from the time-series file and when calculated from the complete transmission history). Further details of model validation are in S1 Appendix.

## Model parameterisation and model deployment

**Running the model.**   The model uses a timestep of 1/48 of a year (a metric week) and ignores leap years.

Random seeds are completely specified and recorded; any model output is thus completely reproducible.

**Baseline population and initial conditions.**   Simulations start in 1900 to allow extensive time for population demographics and the sexual partnership network to stabilize prior to the introduction of HIV (as adult individuals in 1900 start with zero partners; see S1 Appendix for further details).

The introduction of HIV is simulated in the population through seeding a small proportion of the population. This seeding process is repeated annually over multiple years so that epidemics with $R_0$ just above 1, which may potentially reproduce the epidemic trend if they do not die out due to stochastic variation in the model, are not excessively disfavoured, and to allow for the possibility that in reality HIV may have entered a population multiple times before becoming endemic.

**Model parameterization and calibration.**   When deployed, models are typically paired with one of many different calibration approaches. To illustrate the coupling of PopART-IBM with a calibration framework we describe parameterization of the model with a Bayesian framework, using an adaptive population Monte Carlo Approximate Bayesian Computation algorithm (APMC-ABC) [26], implemented via the R library package EasyABC version 1.5 [27]. An ABC framework was adopted because it is flexible, has been applied to many different data types (including phylogenetic data; e.g. [28]), is widely available in a range of packages (minimising human resource and testing time), and is a widely cited framework for pairing with dynamical models (e.g. [29]). While the IBM contains over 350 parameters, only 17 parameters are estimated through the calibration process used in this article (Table 2). These parameters were those where there was limited data in the literature to estimate them.

The APMC-ABC inference is carried out iteratively, comparing simulated summary statistics with observed summary statistics so as to minimize a Euclidean distance measure. In each inference step 2,000 simulations of the model are carried out, and the 50% of the simulations with the lowest distance are retained for the next step. 1,000 additional new parameter sets are then drawn based on an algorithm described in [26], prior to the next step. The APMC-ABC algorithm stops when the proportion of simulations with decreased distance measure is below 5%, with the threshold chosen based on the observation from [26] that $p_{accmin} < 0.1$ *give good results for the criterion of "number of simulations" x $L^2$ (distance measure)*. The resulting distribution of accepted model projections is proportional to the posterior distribution given the data and the structure of the IBM [26]. A full list of the data sources used to parameterize the model in this study, and marginal posterior distributions of parameters after calibration, is given in S1 Appendix, including detailed descriptions of the analyses and checks used for parameters related to sexual behaviour such as partnership duration.

The data sources used in the calibration process illustrated here include the HPTN 071 (PopART) population cohort (PC) survey collected from 2013–2018; three rounds of Zambian Demographic Health Surveys (DHS) in 2002, 2007, and 2013; and data collected by CHiPs

**Table 2. Parameters estimated in the calibration framework.** Uniform priors were used for all parameters with ranges shown in the table.

| Parameter description | Lower range | Upper range | Distribution |
|---|---|---|---|
| Risk assortativity | 0.05 | 0.95 | uniform |
| Baseline probability of a woman having an HIV test in the period before 2006. | 0.1 | 0.2 | uniform |
| Baseline annual probability of a woman having an HIV test in a given year after 2006 | 0.05 | 0.4 | uniform |
| Multiplier for baseline annual probability of a man having an HIV test in a given year compared to a woman. | 0.4 | 1.1 | uniform |
| Per-timestep hazard of transmission from an HIV+ individual to a HIV- partner. | 0.05 | 0.3 | uniform |
| Probability that an individual not in PopART collects CD4 test results and therefore join pre-ART care. | 0.75 | 0.95 | uniform |
| Multiplier of hazard of HIV infection if transmission is from male to female. | 1 | 3 | uniform |
| Proportion of men in the population that are in low risk sexual activity class on entry to the model | 0.3 | 0.6 | uniform |
| Proportion of women in the population that are in low risk sexual activity class on entry to the model | 0.3 | 0.6 | uniform |
| Proportion of men in the medium risk activity class compared to the high sexual risk activity class on entry to the model | 0.5 | 0.99 | uniform |
| Proportion of women in the medium risk activity class compared to the high sexual risk activity class on entry to the model | 0.5 | 0.99 | uniform |
| Multiplier to account for potential over/under-reporting of number of sexual partners. | 0.625 | 5 | uniform |
| Log of the multiplier for the initial proportion of individuals that are infected (stratified by age and sex) during the HIV seeding process. | 0 | 2 | uniform |
| Mean time delay (in years), as drawn from an exponential distribution, until starting ART after testing HIV+ for an individual in the background care cascade. | 0.4 | 0.7 | uniform |
| Multiplier to scale parameter for gamma distribution determining the duration of partnerships within a patch. | 1 | 2 | uniform |
| Probability that a woman stays virally suppressed after the initial ART phase | 0.65 | 0.9 | uniform |
| Multiplier for the probability that a man stays virally suppressed after the initial ART phase compared to a women | 0.6 | 1 | uniform |

teams from 2013–2017 as part of the HPTN 071 (PopART) trial intervention in one trial community. We use as summary statistics HIV prevalence (4 rounds of PC surveys, 3 rounds of DHS, and one round of CHiPs data), as well as the proportion of individuals who are aware of their HIV status (3 rounds of CHiPs data), and the proportion of those aware who are currently on ART (3 rounds of CHiPs data), and the proportion of people living with HIV (PLHIV) who are virally suppressed (which was measured in the third round of the PC survey). Each of these statistics is stratified by age group and sex. Thus there are a total of 248 data points to which the model is calibrated. That is, 116 summary statistics associated with HIV prevalence (6 age groups x 2 sexes x 4 PC rounds + 7 age groups in women x 3 DHS rounds + 9 age groups for men in 3 DHS rounds + 10 age groups x 2 sexes x CHiPs R3) + 60 summary statistics on the proportion of those aware of their HIV status among PLHIV (10 age groups x 2 sexes x 3 CHiPs rounds) + 60 summary statistics on the proportion of individuals on ART (10 age groups x 2 sexes x 3 CHiPs rounds) + 12 summary statistics on viral suppression (6 age groups x 2 sexes x 1 PC round).

During the calibration stage we run the model including the 'intervention' services, since we are calibrating to data including timepoints during the trial when the trial intervention was taking place, generating calibrated parameter sets. In the results we also present model projections from a counterfactual scenario, using these calibrated parameter sets but with the 'intervention' services switched off. This represents a status quo scenario, without PopART-related CHiPs and associated intervention activities, but with HIV testing and antiretroviral therapy available following national guidelines.

**Measuring computational efficiency.** We show two measures of computational efficiency: runtime of the model and the memory used during a run. In an IBM, runtime and memory usage will both depend on population size, and potentially the number of PLHIV. We compiled the model using the Intel icc compiler with '-03' optimization. To measure runtime and memory usage single parameter set was chosen at random from the posterior following

calibration via APMC-ABC, and it was used to generate new parameter sets by keeping all parameters fixed apart from the number of initially seeded HIV infections and the size of the adult population at the beginning of the simulation, which were varied across a grid. The model was run with two patches from 1900 until 2020, and we use the number of adults aged 14 and above in patch 0 in 2020 as the measure of population size. Since runs with very low HIV prevalence run more quickly, due to the reduced number of serodiscordant partnerships, we only include runs where HIV prevalence was at least 10% in 2020. Writing to disk can be a computationally time-expensive task and the model therefore includes the ability to customise which output files are required to be written to disk. Computational experiments measuring runtime and memory usage of the model were performed with a typical set of model output being written to disk (i.e. weekly and annual summaries of population and HIV-related indicators). During calibration the model may be run with only essential indicators used in calibration being written to disk (adjusted using a flag in the compilation of the model) and as such the runtime may be much faster.

Simulations were run on the Oxford BMRC cluster, comprised of a mixture of Intel Ivy-Bridge E5-2650v2 @2.6GHz and Intel Xeon Gold 6126 @2.6GHz processors, with 16GB memory per core and threading disabled. Runtime was measured using the 'real' output from the Linux time command, which measures wall-clock time. Memory usage was measured via the Linux ps command to record the actual memory usage (resident set size) of the model every five seconds, and we computed the maximum memory used over the run. The calibration was run across 12 cores on Oxford's BMRC cluster.

## Results

### Numerical efficiency

PopART-IBM is computationally fast, modelling a 120 year demographic and epidemic history in two linked communities containing around 50,000 adults each in under 30 seconds, and

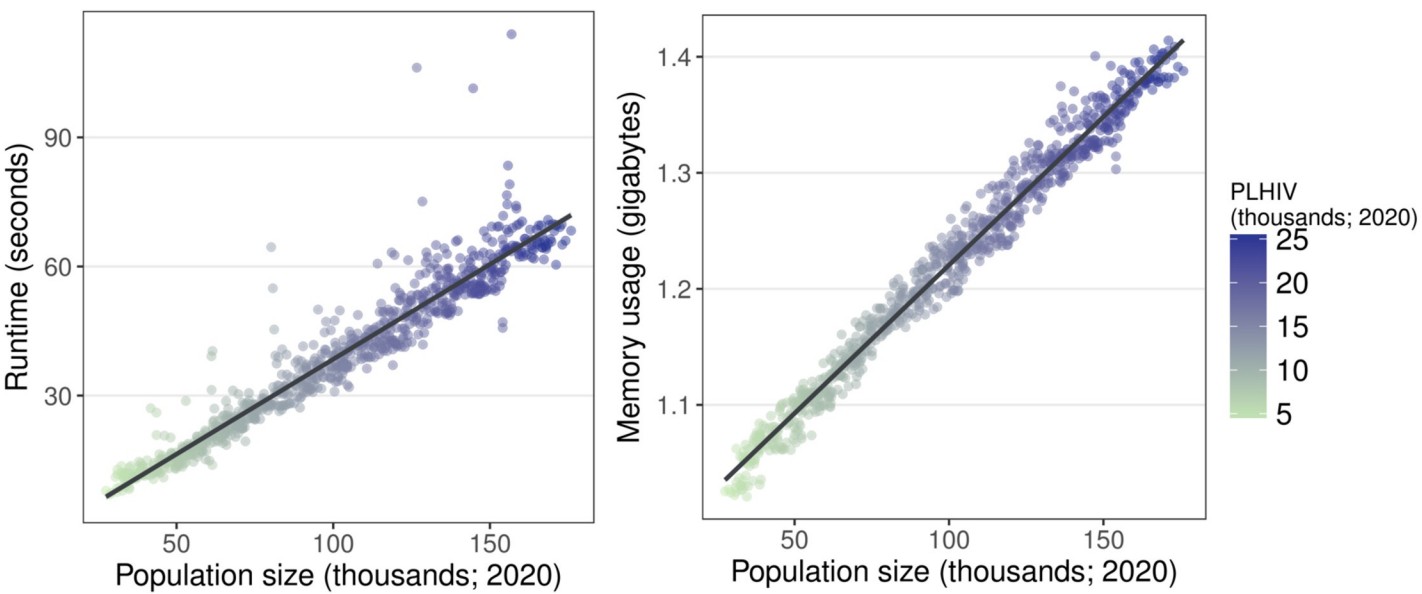

**Fig 3.** Runtime (left) and memory usage (right) of PopART-IBM for a single randomly chosen calibrated parameter set run for 120 years. Population size and PLHIV were measured in one patch at the beginning of 2020. The model is writing a typical set of files to disk in this experiment, including a time series of HIV prevalence, incidence, ART usage, etc. Shorter runtimes can be enabled during calibration by only writing essential indicators used within the calibration process.

 

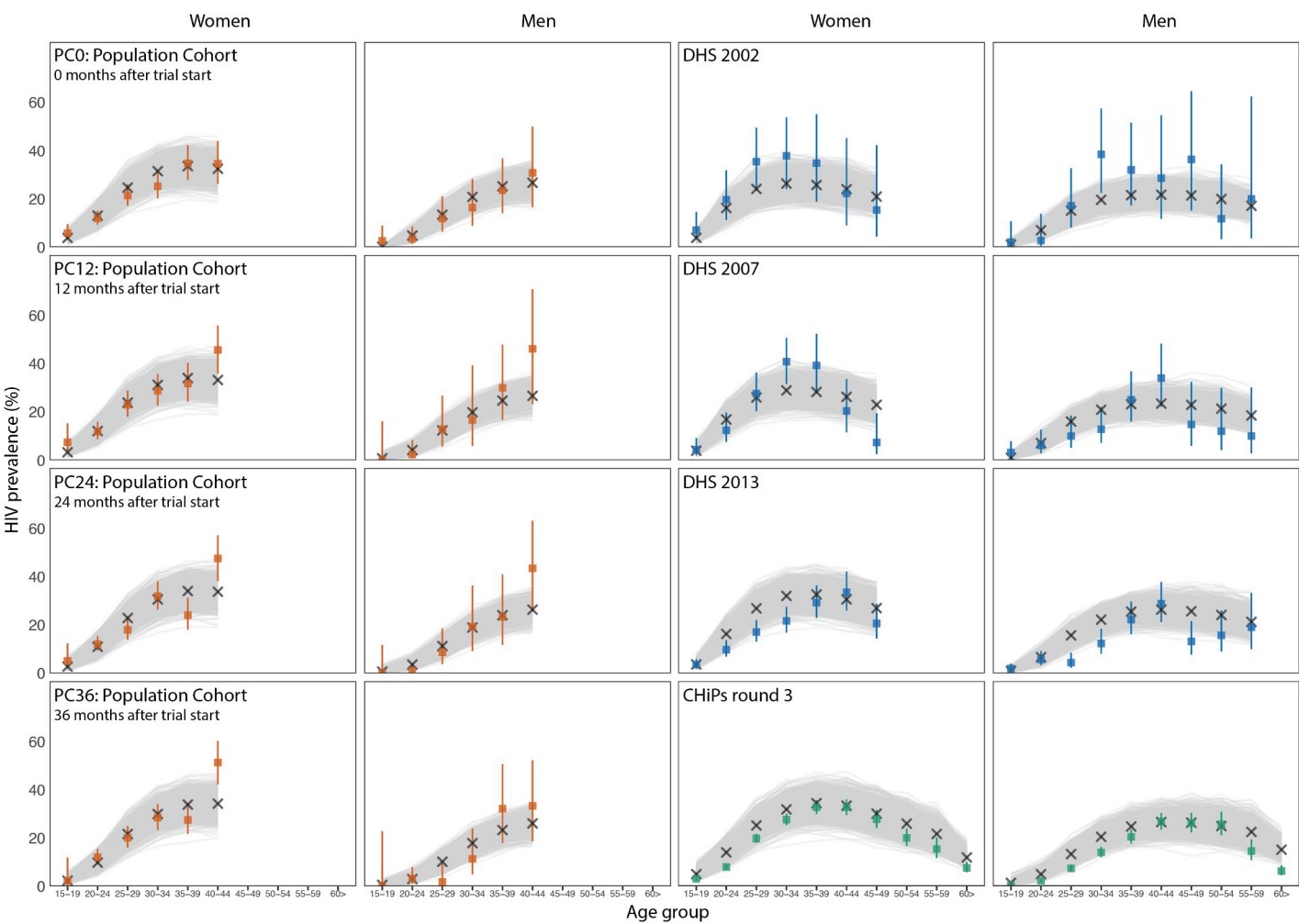

**Fig 4. Calibration of PopART-IBM to PC, DHS and CHiPs datasets for women (columns 1 and 3) and men (columns 2 and 4).** For all graphs, the x-axis shows 5 year age groups, starting at 15–19 years. Panel shows calibration to HIV prevalence in PC0-PC36 (columns 1–2); HIV prevalence from 2002, 2007 and 2013 Zambia DHS surveys (top 3 rows, columns 3–4); HIV prevalence from CHiPs round 3 (bottom row, columns 3–4). In each graph the coloured bars represent the data used for calibration (observed proportions in each subgroup, with 95% confidence intervals), with corresponding colour in the Gantt chart in Fig 5; the grey lines represent the outputs from the 1,000 calibrated model runs; and the crosses represent the best model fit output.

requiring under 2 gigabytes of memory to do so. Fig 3 shows how model computational efficiency varies with the size of the community. Both runtime and memory usage increase linearly as the size of the community increases.

## Example calibration

The APMC-ABC algorithm successfully completed calibrating to the 248 data points after 130 steps, requiring 131,000 simulations. Model fits for 1,000 accepted runs are shown in Figs 4 and 5. The model calibrates well to the majority of the data, and importantly captures overall trends by age and sex for these indicators. Though most fits are good, the model tends to predict a lower proportion of people on ART than self-reported being on ART, however the model predict viral suppression well. These indicators are not self-consistent, reflecting the reality of collecting data from different sources and with different methods: ART coverage here is as reported by the PopART intervention teams working in the general population

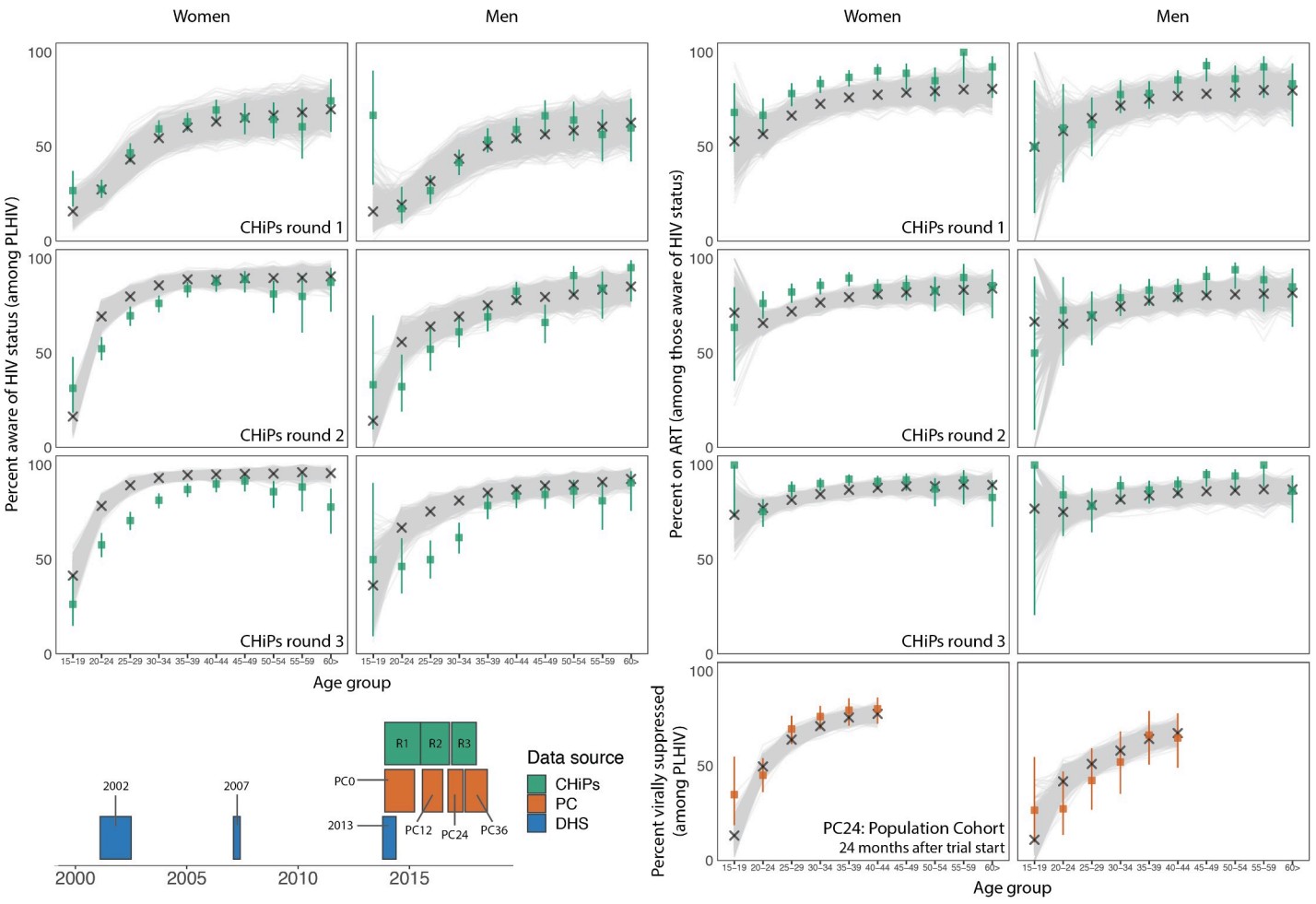

**Fig 5. Calibration of PopART-IBM to PC and CHiPs datasets for women (columns 1 and 3) and men (columns 2 and 4). For all graphs, the x-axis shows 5 year age groups, starting at 15–19 years.** Panel shows percentage of PLHIV who are aware of status from CHiPs rounds 1–3 (columns 1–2), percentage of PLHIV aware of status who are on ART (top 3 rows, columns 3–4); viral suppression amongst all PLHIV from PC24 (bottom row, columns 3–4); and a Gantt chart showing the times that different data sources were collected (bottom left). In each graph the coloured bars represent the data used for calibration (observed proportions in each subgroup, with 95% confidence intervals), with corresponding colour in the Gantt chart; the grey lines represent the outputs from the 1,000 calibrated model runs; and the crosses represent the best model fit output.

(CHiPs teams), whereas viral load suppression is based on samples collected amongst research participants in a population cohort (PC). We did not weight the data; in this instance the model provided a better fit to PC viral load suppression data than to CHiPs data on self-reported ART use. However, data on medication adherence has been documented as being biased upwards (e.g.[30]) so the model estimate of ART usage implicitly accounts for self-reporting bias in this instance.

To inform the history of the epidemic, we used regional survey data collected by the DHS predating the PopART trial. Model outputs after calibration match more closely to HIV prevalence from PC and CHiPs datasets than from DHS data, which may be due to the larger number of data points (both additional age groups and ART cascade outcomes) and sample size of the CHiPs data compared to the regional Zambian DHS data.

On the Oxford BMRC cluster using 12 computer nodes the calibration framework for a single community took approximately 37 hours with a stopping criterion of 5% (as above) (or approximately 5 hours, 8 hours, 14 hours, 70 hours with a stopping rule of 40%, 20%, 10%, and

1% respectively). We caution against generalising compute times for the whole calibration process as compute time may vary dramatically depending on the size of the cluster, parameterisation of the model, number of parameters being varied in the calibration, variability of the data used for fitting (such that a larger area of the parameter space may provide comparably "good" fits), structure of the model, and the choice of output files that are written to disk within each individual simulation (writing to disk is expensive from the point-of-view of computational runtime). During the calibration only essential indicators needed for calibration were written to disk in each simulation (i.e. only the simulated summary statistics). Furthermore, because of the exploratory nature of searching parameter space, some simulations during calibration are much faster than that shown in Fig 3 because the choice of parameters lead to stochastic extinction of HIV (and therefore much faster simulation runtimes).

## Epidemic projections under the status quo

Fig 6 shows projections over time of key model outputs for a scenario without PopART-related CHiPs and associated intervention activities, but with HIV testing and antiretroviral therapy available following national guidelines. The model predicts that per-capita HIV incidence is declining at present, and will continue to decline more slowly until 2030, when it plateaus around 1.02 per 100 person years (2.5% and 97.5% model output quantiles, MOQ 0.665–1.40). The proportion of PLHIV on ART has increased over time, in part driven by changes in ART eligibility over time, and is projected to continue increasing beyond 2030, but only to 59.2% of PLHIV. HIV prevalence has remained roughly stable since 2010, and is 15.1% (MOQ: 11.6–18.4%) in 2030, suggesting that the declines in HIV incidence are offset by PLHIV surviving longer due to ART.

On the right of Fig 6 we see that, as the population in this community is growing, the number of PLHIV is increasing even though HIV prevalence has stabilised, and will grow by 58.5% (MOQ 48.5–67.0%) between 2020 and 2030. The number of people on ART is also increasing almost linearly. Finally, while the number of new cases has declined slightly during the period 2010–2020, it will start to increase again as the proportion of PLHIV on ART plateaus, and by 2030 there will be 18.9% (MOQ 0.0–36.8%) more incident cases than in 2020. While there is substantial uncertainty in the calibrated model fits, these trends remain present across all runs. Simulations are closely determined by the uptake of ART over time; here we use a simple continuous logistic function for the rates of testing and linkage over time, and these scenarios miss UN targets for ending AIDS in 2020 and 2030. Other scenarios representing more substantial ramp up of ART coverage will be explored in further work.

## Epidemic dynamics

Fig 7 shows a simulated transmission tree, created from a single model run. Time is displayed along the x-axis, and nodes (circles) represent individuals, located at the time of infection, with colour denoting sex. Transmission pairs are joined by an edge (line), with thicker lines representing when transmission occurred during early HIV infection, which can particularly be seen in the clusters where several transmission events occurred during a short timescale. This pattern shows that the dynamics of this model, in well calibrated runs, displays dynamics not easily described by a deterministic model. Namely, the model predicts two tempi of transmission, with slow low-branching transmission chains, interspersed with rapid outbreaks amongst high risk individuals connected by multiple concurrent partnerships.

## Discussion

The PopART-IBM is a fast, computationally efficient model of HIV transmission, simulating 50 years of an HIV epidemic in a typical, high-prevalence community of approximately

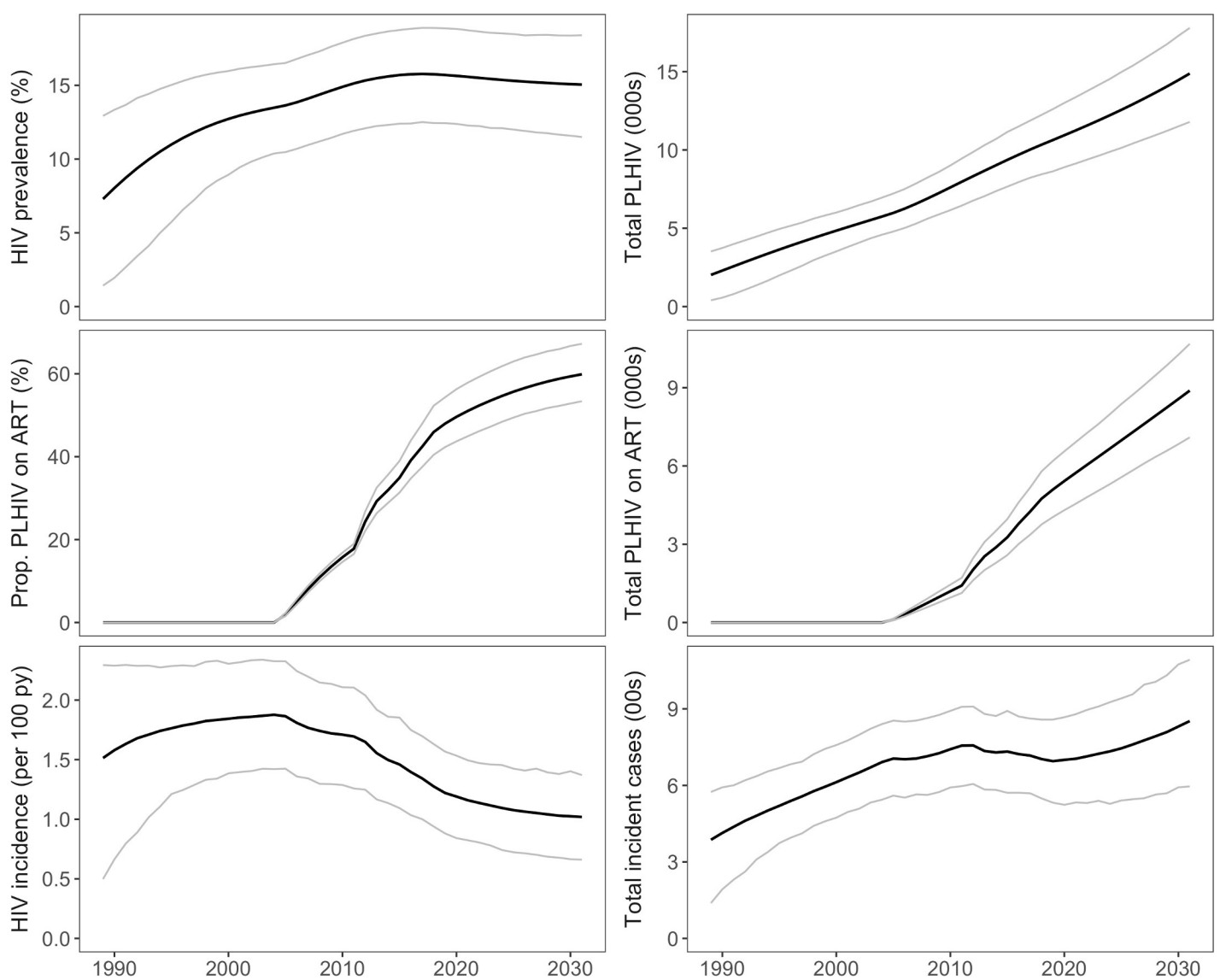

**Fig 6. Projected outputs over time from 1,000 calibrated runs from PopART-IBM for a counterfactual scenario with PopART-related intervention HIV testing, linkage to care and VMMC switched off.** Left column shows normalized or percentage outputs, such as HIV incidence and prevalence, while right column shows absolute outputs, such as total number of PLHIV. Black line shows median and grey lines show 2.5% and 97.5% quantiles from model output. Clockwise from top left: HIV prevalence over time; total number of people living with HIV/AIDS over time (thousands); total people living with HIV/AIDS who are on ART over time (thousands); incident cases per year; HIV incidence rate; percentage of PLHIV who are on ART. py = person years.

100,000 people in under a minute on a standard laptop. It is capable of exploring hundreds of thousands of parameter combinations on a standard computer cluster. In this article we have demonstrated calibration to a single representative community in the HPTN 071 (PopART) trial, and presented some of the detailed outputs that can be generated easily with such an individual-based model. The computational efficiency of the model scales linearly, meaning that the model can simulate an epidemic across a country of 50 million people in about a day on existing, specialist, high-performance nodes.

The model projections we show here highlight that while per-capita outputs such as HIV incidence and prevalence may stabilise, under the status quo, the growing population leads to increases in the number of new infections per year and the number of PLHIV on ART, and

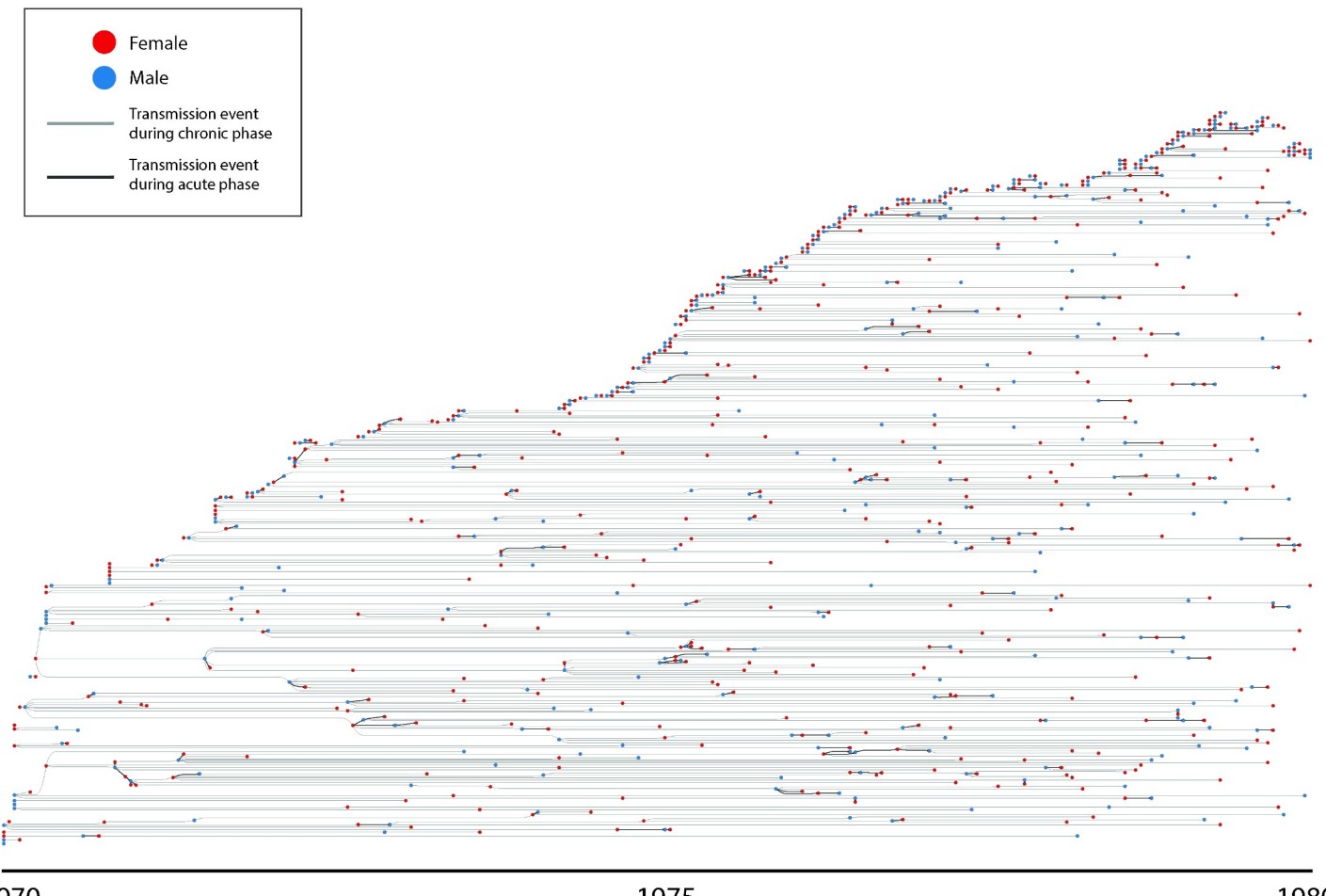

**Fig 7. Transmission tree from a single calibrated run of the PopART-IBM over the period 1970–1980.** Nodes (circles) represent HIV infected individuals, and are coloured by sex. Edges (straight lines) represent transmission events, with infectors on the left and infectees on the right. Thicker lines represent where infections occurred when the infector was in the early stages of HIV infection.

the latter will have consequences for healthcare resources. The UN targets for 2030 aim to 'end AIDS' and send the epidemic into sustained decline; continuation of the status quo will not just fail to achieve these ambitions, but may in fact result in an epidemic that is continuing to grow in some high prevalence settings. In future work, we will explore further scenarios in the light of the findings of the HPTN 071 (PopART) trial, including cost-effectiveness.

We have also shown a transmission tree generated by the model, providing a different perspective on HIV transmission in the model that explicitly demonstrates the clustering of transmission in a way that is not possible through deterministic models, and provides outputs for comparison with phylogenetic methods [31].

The model generally calibrates very well to data from three different sources (DHS, CHiPs and population cohort), coming from eight time points. Using these data we calibrate to three different outcomes (HIV prevalence, proportion of PLHIV on ART, and viral suppression) stratified by age and sex. While the quality of the fit to early rounds of DHS data is somewhat lower, the DHS data itself has high variability both within and between rounds that make it challenging to fit to. By fitting to multiple data sources and time points we expect to reduce the impact of inconsistencies in the data used for calibration. In addition, model predictions for

the proportion of PLHIV on ART are somewhat lower than the data, though this may be expected as the data are from self-reported measures whereas the model assumes these are the "true" proportion on ART among those aware of their status. Other research has highlighted that self-reported ART usage can overestimate ART usage [32]. The fact that we also calibrate to viral load gives us more confidence that the model's lower projected ART status is not unduly affecting our findings more generally.

While a number of individual-based models of HIV transmission exist (e.g. MicroCOSM [33,34], EMOD [35,36], STDSIM [37,38], HIV Synthesis Transmission model [39,40]), in a systematic review of individual-based models Abuelezam et al. [41] noted that "the rigor in reporting of assumptions, methods, and calibration of individual-based models focused on HIV transmission and prevention varies greatly." Porgo et al. [42] similarly state "While individual-based models can provide more realistic representations of a system, they can be difficult to parameterize because they require much more detailed knowledge, or assumptions, of how variables interact. The stochastic nature of these models makes them computationally intensive and challenging to calibrate". In this study we have shown that the PopART-IBM can be parameterized using local survey data, including age-mixing and other partnership-related parameters, and successfully calibrated in a Bayesian framework to large quantities of age and sex-specific data on HIV prevalence, uptake of HIV testing and antiretroviral therapy, and viral suppression, coming from different sources.

Limitations of the model include structural choices of what to include and what not to include. We took a parsimonious approach based on data availability and prior experience of determinants of population dynamics, but we cannot rule out that including factors such as migration, ART regimen and drug resistance, and spread in key populations would result in different projections. With declining incidence, key populations are likely to play a bigger role in the future and modelling these dynamics will require additions to the model, a topic of current research.

Another limitation is that our code was developed with numerical efficiency in mind; as a result it may be more difficult to adapt to different settings and different assumptions than other IBMs. We reasoned that numerical efficiency was essential for calibration and validation.

Work is ongoing to use PopART-IBM in the context of the HPTN 071 (PopART) trial, to provide additional insight into the trial results [18], including to better understand the effects of age and sexual risk on transmission in the trial [43], and to generate cost-effectiveness estimates.

To test the validity of the model, predictions of the model were publicly logged prior to the unblinding of the trial, and work is underway to examine what did or did not work in the modelling, allowing greater model validation [44], and providing recommendations for how models can better be used to inform and predict HIV trials. Outputs from the model are being combined with results from phylogenetic analyses in the same HPTN 071 (PopART) trial communities in Zambia to cross-validate the findings of each method into patterns of HIV transmission. Finally, the model is being adapted with the addition of pre-exposure prophylaxis for use in Manicaland, Zimbabwe.

PopART-IBM is a computationally efficient stochastic individual-based model of HIV transmission, calibrated within a Bayesian framework, that can be flexibly customized to different settings and interventions, and provide a range of key outputs. By making the model open source we hope that this can become a tool that is of use to HIV epidemiologists and public health experts more widely. Robust calibration to ever richer data sources may provide new granular insights into epidemic dynamics, and help design, support and evaluate new HIV prevention efforts.

## Supporting information

**S1 Appendix.** Fig A. Annual mortality for men in Zambia by age group. Fig B. Annual mortality for women in Zambia by age group. Fig C. Annual mortality for men in South Africa by age group. Fig D. Annual mortality for women in South Africa by age group. Fig E. Definition of sexual activity levels in PopART-IBM. Fig F. HIV prevalence by sex, sexual activity level and age group for Zambia and South Africa in PC0 data. Fig G. Rate of partnership formation inside and outside the community, by age, activity class and sex for Zambia. Fig H. Rate of partnership formation inside and outside the community, by age, activity class and sex for South Africa. Fig I. Observed and fitted duration of partnerships so far for Zambia. Fig J. Observed and fitted duration of partnerships so far for South Africa. Fig K. Schematic of the HIV care cascade in the model. Fig L. Age distribution by sex for 4 randomly chosen calibrated IBM runs in Zambia. Fig M. Population size over time for 4 randomly chosen calibrated IBM runs in Zambia. Fig N. Percentage of the population who are in the low, medium and high sexual activity level groups over time, for the 4 randomly chosen calibrated runs. Fig O. Histogram showing the distribution of time from infection to AIDS death for individuals dying by 2004. Fig P. Histogram showing the distribution of time from infection to AIDS death for individuals dying in one calibrated run, by set-point viral load category. Fig Q. Distribution of PLHIV in the ART cascade over time for the four randomly chosen calibrated runs. Fig R. Changes in transitions between ART cascade states of PLHIV over time for one randomly chosen calibrated run. Fig S. Mean number of lifetime, current, and new partners of individuals currently alive in the simulation, plotted over time by sexual activity level for the 4 randomly chosen calibrated runs. Fig T. HIV prevalence for 200 uncalibrated runs. Fig U. Marginal posterior distributions for the 17 calibrated parameters. Table A. List of epidemiological characteristics of each adult individual stored by PopART-IBM. Table B. Parameters related to initialization of the population in PopART-IBM. Table C. Fertility rate for Zambia over time. Table D. Fertility rate for South Africa over time. Table E. Years for which UNPD mortality estimate is used in regression model. Table F. Country, sex and age-group specific mortality parameters. Table G. Partnership-related parameters used in PopART-IBM. Table H. Proportion of population in each sexual activity level by sex. Table I. Age mixing matrix for men in Zambia. Table J. Age mixing matrix for women in Zambia. Table K. Age mixing matrix for men in South Africa. Table L Age mixing matrix for women in South Africa. Table M. Parameters related to initializing HIV in PopART-IBM. Table N HIV transmission-related parameters used in PopART-IBM. Table O HIV progression-related parameters used in PopART-IBM. Table P. Cascade-related parameters used in PopART-IBM. Table Q Parameters for time to ART initiation after receiving a positive HIV test result from CHiPs in PopART-IBM. Table R Circumcision-related parameters used in PopART-IBM.
(PDF)

## Acknowledgments

We are grateful to all members of the HPTN 071 (PopART) Study Team and to the study participants and their communities for their contributions to this research. We also thank Lucie Abeler-Dorner for helpful discussions and comments on the manuscript.

## Author Contributions

**Conceptualization:** Michael Pickles, Anne Cori, William J. M. Probert, Rafael Sauter, Christophe Fraser.

**Data curation:** Helen Ayles, Peter Bock, Deborah Donnell, Ethan Wilson, Sian Floyd.

**Formal analysis:** Michael Pickles, Anne Cori, William J. M. Probert, Rafael Sauter, Robert Hinch.

**Funding acquisition:** Sarah Fidler, Helen Ayles, Peter Bock, Richard J. Hayes, Christophe Fraser.

**Investigation:** Michael Pickles, Anne Cori, William J. M. Probert, Rafael Sauter.

**Methodology:** Michael Pickles, Anne Cori, William J. M. Probert, Rafael Sauter, Christophe Fraser.

**Resources:** Helen Ayles, Peter Bock, Estelle Piwowar-Manning.

**Software:** Michael Pickles, Anne Cori, William J. M. Probert, Rafael Sauter, Robert Hinch, Christophe Fraser.

**Supervision:** Christophe Fraser.

**Validation:** Michael Pickles, Anne Cori, William J. M. Probert, Rafael Sauter, Robert Hinch.

**Visualization:** Michael Pickles, Anne Cori, William J. M. Probert, Rafael Sauter, Robert Hinch.

**Writing – original draft:** Michael Pickles, Anne Cori, William J. M. Probert, Rafael Sauter, Christophe Fraser.

**Writing – review & editing:** Michael Pickles, Anne Cori, William J. M. Probert, Rafael Sauter, Robert Hinch, Sarah Fidler, Helen Ayles, Peter Bock, Deborah Donnell, Ethan Wilson, Estelle Piwowar-Manning, Sian Floyd, Richard J. Hayes, Christophe Fraser.

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
