## [Decision Letter · Decision Letter 0]

3 Nov 2020

Dear Dr. Pickles,

Thank you very much for submitting your manuscript "PopART-IBM, a highly efficient stochastic individual-based simulation model of generalised HIV epidemics developed in the context of the HPTN 071 (PopART) trial" for consideration at PLOS Computational Biology.

As with all papers reviewed by the journal, your manuscript was reviewed by members of the editorial board and by several independent reviewers. In light of the reviews (below this email), we would like to invite the resubmission of a significantly-revised version that takes into account the reviewers' comments.

All three reviewers did appreciate the manuscript, however raised some concerns regarding the presentation as well as some model assumptions. In the revised version of the manuscript, the authors should in particular try to illustrate and explain the main aspects of the model in the manuscript rather than moving all technical aspects to the supplementary material. Moreover, the study aim, some model assumptions and limitations of the model should be discussed in more detail. In addition, the authors should guarantee that re-running their code from the gitlab repository is possible.

We cannot make any decision about publication until we have seen the revised manuscript and your response to the reviewers' comments. Your revised manuscript is also likely to be sent to reviewers for further evaluation.

Sincerely,

Katharina Kusejko

Guest Editor

PLOS Computational Biology

Rob De Boer

Deputy Editor

PLOS Computational Biology

Reviewer's Responses to Questions

**Comments to the Authors:**

Reviewer #1: Attached

Reviewer #2: Attached

Reviewer #3: Attached

**Have all data underlying the figures and results presented in the manuscript been provided?**

Reviewer #1: Yes

Reviewer #2: Yes

Reviewer #3: Yes

PLOS authors have the option to publish the peer review history of their article (what does this mean?). If published, this will include your full peer review and any attached files.

Reviewer #1: No

Reviewer #2: No

Reviewer #3: No
---

## [Decision Letter · Decision Letter 1]

14 Jun 2021

Dear Michael Pickles,

Thank you very much for submitting your manuscript "PopART-IBM, a highly efficient stochastic individual-based simulation model of generalised HIV epidemics developed in the context of the HPTN 071 (PopART) trial" for consideration at PLOS Computational Biology. As with all papers reviewed by the journal, your manuscript was reviewed by members of the editorial board and by several independent reviewers. The reviewers appreciated the attention to an important topic. Based on the reviews, we are likely to accept this manuscript for publication, providing that you modify the manuscript according to the review recommendations.

Sincerely,

Katharina Kusejko

Guest Editor

PLOS Computational Biology

Rob De Boer

Deputy Editor

PLOS Computational Biology

Reviewer's Responses to Questions

**Comments to the Authors:**

Reviewer #1: The manuscript is much improved, readable and flowing.

Comment:

Is it necessary to have both the abstract and author summary?

Reviewer #2: The manuscript has been greatly improved since the first submission and is ripe for publication!

I have already noted this to the editor, but for the relief of the authors I'll note this here, too (especially if it is brought up again by my fellow reviewers): Upon review of the second submission, I noticed that downloading the individual figure files produced high quality images, which the authors were quite certain they had submitted from the start. However, within the compiled pdf of the whole paper, as provided for the reviewers, the figures again appear very poor and with low resolution. Thus, it seems that the compilation process used by PLOS Computational Biology to yield a pdf for review is messing with the quality of the figures within the manuscript. This should be noted for future submissions and of course not held against the authors.

Reviewer #3: Main comment:

The authors replied with full details to almost all the concerns/questions that were raised. Only one concern is remaining, which is related to the comment 2.17. To me, the Figure 5 (which was Figure 4 in the previous version) still does not bring anything that justifies keeping it in the main text. Moreover, as the focus of this manuscript is on the model (and not the interpretation of the results), describing the dynamic of HIV as predicted by the model for a single counterfactual scenario (lines 493-502) do not really help the reader understand the features of the model.

Therefore, I would consider either presenting different counterfactual scenarios (e.g. the 3 arms of the study, in order to show what the model is able to compare, even if I understand that the exhaustive comparison of these arms will be done in a subsequent paper) or removing Figure 5 and the related paragraphs describing the results.

Minor comments:

There is a small typo (Figure55 should be Figure5).

There are two Tables 2. The first one should be Table 1.

The Table 2 (the second table describing the parameters) really helps understand which parameters are involved in the model. However, I would remove the “technical” names of the parameters (first row), as these names are only used in the model script.

**Have the authors made all data and (if applicable) computational code underlying the findings in their manuscript fully available?**

Reviewer #1: Yes

Reviewer #2: Yes

Reviewer #3: Yes

PLOS authors have the option to publish the peer review history of their article (what does this mean?). If published, this will include your full peer review and any attached files.

Reviewer #1: No

Reviewer #2: No

Reviewer #3: No

Figure Files:

Data Requirements:

Reproducibility:

References:

---

## [Editor Report · Decision Letter 2]

22 Jul 2021

Dear Dr. Pickles,

We are pleased to inform you that your manuscript 'PopART-IBM, a highly efficient stochastic individual-based simulation model of generalised HIV epidemics developed in the context of the HPTN 071 (PopART) trial' has been provisionally accepted for publication in PLOS Computational Biology.

Best regards,

Katharina Kusejko

Guest Editor

PLOS Computational Biology

Rob De Boer

Deputy Editor

PLOS Computational Biology

---

## [Editor Report · Acceptance letter]

27 Aug 2021

PCOMPBIOL-D-20-01554R2 

PopART-IBM, a highly efficient stochastic individual-based simulation model of generalised HIV epidemics developed in the context of the HPTN 071 (PopART) trial

Dear Dr Pickles,

I am pleased to inform you that your manuscript has been formally accepted for publication in PLOS Computational Biology. Your manuscript is now with our production department and you will be notified of the publication date in due course.

With kind regards,

Andrea Szabo
